# Comparison of Surgical Outcomes of Robotic versus Conventional Laparoscopic Hysterectomy of Large Uterus with Gynecologic Benign Disease

**DOI:** 10.3390/jpm12122042

**Published:** 2022-12-10

**Authors:** Soo Young Jeong, Kyoungseon Kim, Ji Won Ryu, Jieum Cha, Sung Taek Park, Sung Ho Park

**Affiliations:** Department of Obstetrics and Gynecology, Kangnam Sacred-Heart Hospital, Hallym University Medical Center, Hallym University College of Medicine, Seoul 07441, Republic of Korea

**Keywords:** robotic hysterectomy, conventional laparoscopic hysterectomy, surgical outcome

## Abstract

Hysterectomy is commonly performed for benign gynecological diseases. Minimally invasive surgical approaches offer several advantages. Unfortunately, few studies have compared the outcomes of different types of minimally invasive surgeries. Therefore, this study aimed to compare the surgical outcomes of robotic hysterectomy (RH) and conventional laparoscopic hysterectomy (CLH) in benign gynecologic diseases. We performed a retrospective cohort study at a single center between January 2014 and July 2022. A total of 397 patients (RH: 197 and CLH: 200) who underwent minimally invasive hysterectomy for benign diseases with uterine size exceeding 250 g were enrolled, and factors related to the surgical outcomes were compared. The median age was 46 (range, 35–74) years, and the median uterine weight was 400 (range, 250–2720) g. There were no significant differences between the two groups regarding age, body mass index, uterine weight, hospital stay, estimated blood loss, or operating time. Intraoperative and postoperative complication rates were not significantly different between the two groups. RH was not inferior to CLH in terms of perioperative and immediate postoperative outcomes in our study.

## 1. Introduction

Hysterectomy is commonly performed for benign gynecologic diseases such as uterine myoma, adenomyosis, endometriosis, genital prolapse, pelvic pain, and other symptoms associated with pelvic organs. [1]. The prevalence of hysterectomy in the reproductive years (ages 18–44 years) has reached about 18% and that of remaining ages reached about 48% in the United States. These prevalence rates showed a consistency through 1997–2005 [2]. An extensive study about unadjusted all-age hysterectomy prevalence using the Behavioral Risk Factor Surveillance System, providing national representative trends, revealed that it ranged from 21.4% in 2006 to 21.1% in 2016 [3]. Similar trends were noted in the Republic of Korea. Although overall rates of hysterectomy have been decreasing, 329.6 of 100,000 women in Korea underwent hysterectomy in 2010, according to health data from the Organization for Economic Cooperation and Development [4].

For hysterectomy, surgeons usually choose a surgical approach based on the clinical circumstances or personal preferences. Abdominal or vaginal hysterectomy has traditionally been performed. However, minimally invasive approaches including conventional laparoscopy, 3D laparoscopy, and robotic surgery are being increasingly applied in benign gynecological surgery [5,6]. Of all the hysterectomies in USA in 2003, the most common method was abdominal (66.1%), followed by vaginal (21.8%) and laparoscopic hysterectomy (11.8%) for benign disease [7]. By 2009, the tendency had shifted toward laparoscopic (20.4%) and robotic surgery (4.5%) [8]. However, in Korea, the change was rapid; laparoscopic hysterectomy in benign cause accounted for 52.0% by 2009, with a significantly increasing trend (*p* < 0.001) [9].

This was inevitable for minimally invasive surgical approaches as they offer several important advantages such as less pain, reduced blood loss, shorter duration of hospitalization, faster postoperative recovery, smaller scars, and fewer complications than open methods [10,11]. Therefore, the abdominal hysterectomy rates fell short of minimally invasive surgery in 2012 [12].

Among the minimally invasive surgeries, surgeons have usually chosen the conventional laparoscopic technique for hysterectomy. It requires an endoscopic camera and long instruments, with small abdominal incisions [13]. Since the approval of the da Vinci robotic surgical system by the U.S. Food and Drug Administration in 2005, the number of robotic surgeries has been increasing [14]. The advantages of robotic surgery include three-dimensional (3D) visualization, tremor filtration, higher magnification, telestration, and improved ergonomics by using EndoWrist instruments (Intuitive Surgical Inc., Sunnyvale, CA, USA) that provide freedom of articulation with improved visualization and dissection precision [15,16]. Robotic hysterectomy occupied the highest rate (robotic, 36%; conventional laparoscopic, 31%; abdominal, 24%; and vaginal, 8%) in 2013 among all benign hysterectomies, and its contribution has been increasing rapidly [12]. Therefore, we aimed to compare the surgical outcomes of robotic hysterectomy (RH) and conventional laparoscopic hysterectomy (CLH) in benign gynecologic diseases.

## 2. Materials and Methods

### 2.1. Patient Selection and Data Collection

In this retrospective cohort study, data from women who underwent minimally invasive hysterectomy at Kangnam Sacred-Heart Hospital, Seoul, Korea, between January 2014 and July 2022 were reviewed. As this was a retrospective study, direct written informed consent from patients was not required, as per the ethical guidelines.

The inclusion criteria were (1) patients with a large uterus size over 250 g; (2) who underwent minimally invasive hysterectomy; and (3) who had benign diseases. Patients with gynecological cancers were excluded. A total of 403 patients were enrolled and among these, six patients were excluded because other procedures such as cholecystectomy or breast mass excision were also performed (Figure 1). Finally, 397 patients were included in the study. Of these, 197 women underwent robotic surgery and 200 underwent conventional surgery.

A single gynecologic surgeon (S.H.P.), who has been performing laparoscopic and robotic surgeries for 20 and 8 years, respectively, evaluated all patients enrolled in the study preoperatively and performed all surgeries. He decided the surgical approach according to the patient’s characteristics and clinical parameters. Hence, all surgeries had similar perioperative management and intraoperative strategies. For CLH, a primary 10 mm port was positioned at the umbilicus and an additional two or mostly three 5 mm ports were placed in the suprapubic region and at each side of the abdomen, resulting in a diamond-shaped port placement. For RH, a primary 8 mm robotic port was positioned at the umbilicus. Subsequently two more 8 mm robotic ports were placed lateral to the primary port, each about 8 cm apart. The assistant’s port was placed on the right to one of the additional ports, resulting in a linear port alignment. Except for port placement, almost all of the surgical procedures were carried out in a similar manner including vaginal retrieval of the specimen and intraabdominal stump suture. Even the suture material was manufactured by the same company. For the energy device, Thunderbeat (Olympus, Tokyo, Japan) and Fenestrated bipolar forceps (Intuitive Surgical Inc., Sunnyvale, CA, USA) were utilized.

Data were retrieved from the patients’ electronic medical records. Operative time was defined as the time from skin incision to skin closure, which included docking time for RH. Drop in hemoglobin level was defined as the difference between the preoperative and postoperative hemoglobin levels. Postoperative hemoglobin was measured in the morning following surgery. Postoperative complications were classified according to the Clavien–Dindo classification [17].

### 2.2. Statistical Analyses

Statistical analyses were performed using SPSS version 25.0 (SAS Institute, Cary, NC, USA). The descriptive statistics were reported as median (range) for continuous variables (age, body mass index [BMI], uterus weight, hospital days, estimated blood loss [EBL], operative time, and drop in hemoglobin), and number (percentage) for categorical variables (number of previous operations, uterine disease, pelvic adhesion, conversion to open surgery, intraoperative complications, and postoperative complications). Clinical data were compared using χ2 or Fisher’s exact tests for categorical variables and Student’s t- or Wilcoxon rank-sum tests for continuous variables. Two-sided tests were applied and *p* < 0.05 was considered statistically significant. Values were reported to the thousandths, which were rounded up from the ten-thousandths.

## 3. Results

Consecutive participants (*n* = 403) were screened, but six were excluded because they also underwent non-gynecologic surgery. The final cohort comprised 397 participants: 197 patients in the RH group and 200 patients in the CLH group. The participants’ characteristics are shown in Table 1. The median age was 46 years (range, 35–74), median BMI was 23.6 kg/m^2^ (range, 15.4–42.7), and the median uterus weight was 400 g (range, 250–2720). In RH, the median age was 47 years (range, 36–42), median BMI was 23.4 kg/m^2^ (range, 17.7–42.7), the median weight of the uterus was 430 g (range, 250–2000). In CLH, the median age was 47 years (range, 35–74), median BMI was 24.0 kg/m^2^ (range, 15.4–40.9), and the median weight of the uterus was 363 g (range, 250–2720). There were no significant differences between the two groups in terms of age, BMI, and uterine weight. The two most common indications in both groups were myoma (RH, 91.4%; LH, 94.5%) and adenomyosis (RH, 64.5%; LH, 61.5%). There were no significant inter-group differences in the number of previous operations or pelvic adhesions.

The surgical outcomes are presented in Table 2. The median hospital stay (5 days in both groups), median EBL (RH, 100 mL; LH, 150 mL), and the drop in hemoglobin (1.6 g/dL in both groups) were similar in both groups. Only one case was converted to laparotomy in the CLH group. Intraoperative complications comprised two types: blood transfusion and ureteral injury. Six patients in each group received blood transfusions. Additionally, four patients in the robotic group and one patient in the laparoscopic group had ureteral injury. Postoperative complications were classified using the Clavien–Dindo classification. Urinary retention with catheterization was classified as a grade 1 complication, and there was no significant difference between the two groups (RH, 4.1%; LH, 3.5%, *p* = 0.576). Postoperative transfusion and antibiotic use were classified as grade 2 complications, and there was no significant difference between the two groups (RH, 8.1%; LH, 12.5%, *p* = 0.438).

## 4. Discussion

Minimally invasive surgery has become a tremendously important surgical technique over the past three decades. It was proven with strong evidence that it has better surgical and patient outcomes—a reduction in hospital days, EBL, postoperative pain, postoperative morbidities, and cost—than open approaches [18,19]. Additionally, since the FDA approved the da Vinci robotic surgical system in 2005, the number of RH has been increasing annually. However, the benefits of RH compared to CLH are still being debated [20,21]. This retrospective study demonstrated that RH is not inferior to CLH in terms of operation outcomes.

In two randomized controlled trials, RH was found to require a significantly longer operative time [22,23]. Operating time is influenced by patient-related or surgeon-related factors. Patient-related factors including older age, higher BMI, increased uterine weight, and adhesions can increase the operating time. We established that patient-related factors were not different enough to affect the findings on that matter. Surgeon-related factors including surgical approach, technique, and surgeon expertise can also increase the operating time [24]. In particular, the surgeon’s expertise is the most important contributing factor to operating time [25]. In this study, we excluded all surgeries performed by surgeons other than S.H.P.; therefore, the effect of expertise could be eliminated. Additionally, the exact docking time of RH was not stated in the electronic medical records, thus we calculated the operating time including docking time. All the procedures related to robotic docking were conducted under supervision by S.H.P. Despite including docking time in the operating time, there was no significant difference between RH and CLH groups, thereby indicating that RH is not inferior to CLH in terms of operating time.

Several studies have compared the complication rates of RH and CLH. Most studies have reported no difference in the complication rates between the two groups. In a cohort study of 264,758 women in the U.S., the overall complication rates were similar for RH and CLH (5.5% vs. 5.3%) [26]. However, one study showed a slightly higher complication rate with CLH (17.59% in CLH vs. 9.65% in RH) [27]. Our study found no significant differences between the two groups. Five patients experienced ureteral injury during hysterectomy (four in RH and one in CLH). In all cases, it was confirmed that the ureteral wall was weakened, not perforated or cut; therefore, it was resolved by inserting a double-J catheter into the ureter. Several studies have reported major complications in RH such as respiratory arrest; however, no other major complications occurred after hysterectomy in this study [26,28].

The present study had several limitations. The study design was retrospective, and the groups were not exactly comparable. However, the bias was in the direction of more challenging cases being offered RH. Furthermore, the study was not a randomized study. While our departmental policy is clear and one surgeon decided the surgical approach of all cases, it is difficult to know the exact decision-making process for individual patients. The main strength of this study is that all cases were operated on by a single surgeon (S.H.P.), resulting in operative management such as bowel preparation, the use of prophylactic antibiotics just before the incision of the skin, supplementation of fluid, and the administration of analgesics for pain control being carried out in a similar way. Operations had no important differences including the number of ports, although the port site of RH differed from that of CLH, the intra-operative strategies, and the vaginal retrieval of specimen. Hence, they eliminate any potential effect of the varying skill levels of surgeons and different pre-, intra-, postoperative management that may have otherwise affected the outcomes.

An individual surgeon-conducted study could also be a disadvantage. A study on the surgical proficiency of RH using cumulative summation analysis showed that proficiency was reached after 33 cases [29]. Another study by a single surgeon mentioned that a significantly decreased operative time was obtained after 24–28 cases [30]. The surgeon (S.H.P.) had already dealt with numerous laparoscopic cases before the introduction of robotic surgery at this center. Considering the total number of RH as 197 cases in the present study, those numbers are not to be neglected. Multiple surgeons whose expertise are not significantly different between the RH and CLH based data analysis are required to tackle this problem.

Although this study concludes that RH is not inferior to CLH, it may be difficult for clinicians to determine the operational approach. Studies including the present study are yet to find the appropriate choice of operation for a patient who has a certain clinical circumstance. The surgical difficulties of the cases in this study varied. Factors such as the weight of the uterus (ranging from 250 g to 2,720 g), the presence or location of mass like myomas or other pelvic structures, which may have obscured the operational field of view, and intraoperative degrees of pelvic adhesion were not classified objectively or subjectively for further evaluation. A review study highlighted that there were no clear indications for RH over other minimally invasive surgeries; however, the RH appeared to be non-inferior to CLH in the hands of expert surgeons [31]

A recent study concluded that robot-assisted techniques did not produce significant clinical enhancements compared to similar surgical techniques with identical outcomes, while their costs were much higher [32,33]. Sisters of Charity Hospital of NY, USA conducted a study on the average surgical cost and revealed that RH cost EUR 4,067 compared to EUR 2,151 for the CLH in July 2021 [34]. In a randomized controlled trial, RH was not found to be advantageous for treating conditions when a vaginal approach was feasible (USD 4,579 for vaginal hysterectomy; USD 7,059 for CLH; USD 8,052 for RH) [35]. Even at our institution, RH costs more than 10 times that of CLH. Hence, a surgeon should also take these factors into account when considering the surgical method, especially in centers located in Korea.

## 5. Conclusions

When introducing robotic surgery, there are numerous concerns about the operation being performed at the console rather than adjacent to the patient. However, several studies have highlighted the advantages of robotic surgery, and the results of this study showed no significant difference compared to those of conventional surgery. Nevertheless, large-scale randomized controlled studies are required to confirm these findings.

## Figures and Tables

**Figure 1 jpm-12-02042-f001:**
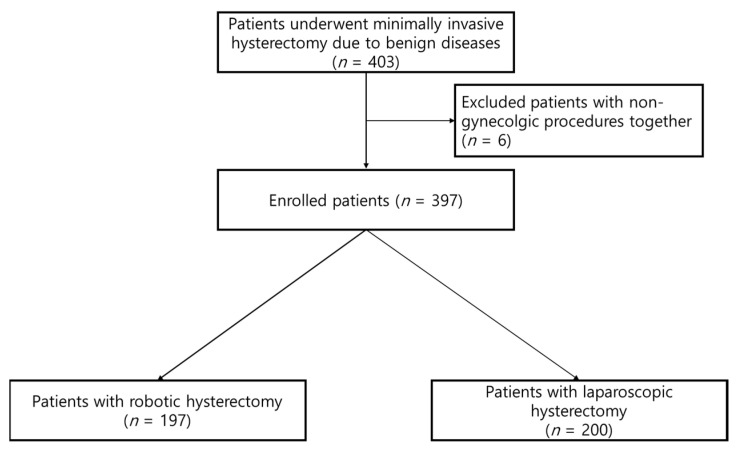
Flowchart for the enrolled patients.

**Table 1 jpm-12-02042-t001:** The patients’ characteristics by the type of hysterectomy.

	Entire Cohort (*n* = 397)	Robotic Hysterectomy (*n* = 197)	Laparoscopic Hysterectomy (*n* = 200)	*p*-Value
Median age (yr)	46 (35–74)	47 (36–62)	47 (35–74)	
Median BMI ^1^ (kg/m^2^)	23.6 (15.4–42.7)	23.4 (17.7–42.7)	24.0 (15.4–40.9)	
Median weight of uterus (g)	400 (250–2720)	430 (250–2000)	363 (250–2720)	
Number of previous operations				0.171
0	213 (53.7%)	111 (56.3%)	102 (51.0%)	
1	92 (23.2%)	49 (24.9%)	43 (21.5%)	
2	76 (19.1%)	29 (14.7%)	47 (23.5%)	
≥3	16 (4.0%)	8 (4.1%)	8 (4.0%)	
Uterus characteristics				
Myoma	369 (92.9%)	180 (91.4%)	189 (94.5%)	0.223
Adenomyosis	250 (63.0%)	127 (64.5%)	123 (61.5%)	0.540
Endometriosis	10 (2.5%)	5 (2.5%)	5 (2.5%)	0.981
Pelvic adhesion	104 (26.2%)	47 (23.9%)	57 (28.5%)	0.293

^1^ BMI, body mass index.

**Table 2 jpm-12-02042-t002:** Surgical outcomes by the surgery type of hysterectomy.

	Entire Cohort (*n* = 397)	Robotic Hysterectomy (*n* = 197)	Laparoscopic Hysterectomy (*n* = 200)	*p*-Value
Hospital days	5 (4–11)	5 (4–11)	5 (5–9)	
EBL (ml) ^1^	100 (10–3000)	100 (20–3000)	150 (10–1000)	
Operative time (min)	120 (60–460)	120 (70–375)	120 (60–460)	
Drop in hemoglobin (g/dL)	1.6 (−1.2–6.5)	1.6 (−0.9–6.5)	1.6 (−1.2–5.3)	
Conversion to laparotomy	1 (0.3%)	0 (0%)	1 (0.5%)	0.320
Intraoperative complication				
Blood transfusion	12 (3.0%)	6 (3.0%)	6 (3.0%)	0.979
Ureter injury	5 (1.3%)	4 (2.0%)	1 (0.5%)	0.172
Postoperative complications ^2^				0.464
I	195 (49.1%)	101 (51.3%)	94 (47.0%)	
II	41 (10.3%)	16 (8.1%)	25 (12.5%)	
III (IIIa, IIIb)	3 (0.8%)	2 (1.0%)	1 (0.5%)	
IV, V	0 (0%)	0 (0%)	0 (0%)	

^1^ EBL, estimated blood loss. ^2^ Postoperative complications were classified as per the Clavien–Dindo classification.

## Data Availability

Not applicable.

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
