# Peer review of "Comparison of Surgical Outcomes of Robotic versus Conventional Laparoscopic Hysterectomy of Large Uterus with Gynecologic Benign Disease"

_jpm, 2022, doi:10.3390/jpm12122042_

Round 1
Reviewer 1 Report
The authors compared the results of robotic vs laparoscopih hysterectomy.
1) why the sample size is not high despite 7 years of period? It is written as high sample size!
Author Response
Thank you for your review point. I checked the number of hysterectomy during 7 years in our hospital one more time and the number was correct. And in my article, it was expressed that the number was not small, compared to other papers. I revised the content of the context.
Reviewer 2 Report
It is a pleasure to referee the article entitled: Surgical outcomes of Robotic versus Conventional laparoscopic hysterectomy of the large uterus.
The study aims to compare the peri and postoperative outcomes of laparoscopic and robotic hysterectomy in case of large uterus.
The article is interesting and has potential, however, I have some comments before publication:
- An extensive editing of English language and style is required. The comprehension is often hindered because of poor English language and too many mistakes. Also the title needs to be modified.
- Lines 27-30: the numbers to express prevalence as they are reported make no sense (…% of 1000?), please revise
- Materials and methods section: including and excluding criteria need to be better specified. Specify the factors for choosing robotic surgery vs laparoscopy. The indication to MIS vs open surgery in this centre should be reported too.
- Lines 115-120: the unit of measurement should immediately follow the value, please revise
- Results section: p value needs should be always reported in the text
- Lines 170-172: add a comment/discussion to the sentence or delete
- Lines 173-175: “The study design was retrospective and the groups were not exactly comparable. However, the bias is in favor of the CLH group; therefore, it does not fundamentally affect the results of this study.” Comment further on this sentence
- The discussion section is too brief and needs to be implemented:
- Comment on the complication rate in this case series vs what is reported in literature
- Further comment factors for ureteral injury, also depending on surgical route
- Recent references and RCT about this topic must be added:
o Paraiso, M.F.R.; Ridgeway, B.; Park, A.J.; Park, A.J.; Jelovsek, J.E.; Barber, M.D.; Falcone, T.; Einarsson, J.I. A randomized trial comparing conventional and robotically assisted total laparoscopic hysterectomy. Am. J. Obstet Gynecol. 2013, 208, 368.e1–368.e3687.
o Sarlos, D.; Kots, L.; Stevanovic, N.; von Felten, S.; Schär, G. Robotic compared with conventional laparoscopic hysterectomy: A randomized controlled trial. Obstet. Gynecol. 2012, 120, 604–611. [CrossRef] 20. Lönnerfors, C.; Reynisson, P.; Persson, J. A randomized trial comparing vaginal and laparoscopic hysterectomy vs robot-assisted hysterectomy. J. Minim. Invasive Gynecol. 2015, 22, 78–86. [CrossRef]
o Capozzi VA, Scarpelli E, Armano G, et al. Update of Robotic Surgery in Benign Gynecological Pathology: Systematic Review. Medicina (Kaunas). 2022;58(4):552. Published 2022 Apr 17. doi:10.3390/medicina58040552
o Lönnerfors, C.; Reynisson, P.; Persson, J. A randomized trial comparing vaginal and laparoscopic hysterectomy vs robot-assisted hysterectomy. J. Minim. Invasive Gynecol. 2015, 22, 78–86.
I will be glad to review the manuscript after minor revision and extensive English revision.
Author Response
Reviewer>
It is a pleasure to referee the article entitled: Surgical outcomes of Robotic versus Conventional laparoscopic hysterectomy of the large uterus.
The study aims to compare the peri and postoperative outcomes of laparoscopic and robotic hysterectomy in case of large uterus.
The article is interesting and has potential, however, I have some comments before publication:
- An extensive editing of English language and style is required. The comprehension is often hindered because of poor English language and too many mistakes. Also the title needs to be modified.
>> Thank you for your comments. English editing will do again.
- Lines 27-30: the numbers to express prevalence as they are reported make no sense (…% of 1000?), please revise
>> Thank you for your comments. I revised the value.
- Materials and methods section: including and excluding criteria need to be better specified. Specify the factors for choosing robotic surgery vs laparoscopy. The indication to MIS vs open surgery in this centre should be reported too.
>> Thank you for your comments. I included the criteria more specific. Also, S.H.P. had decided the surgical approach, so I added the comment in the main text.
- Lines 115-120: the unit of measurement should immediately follow the value, please revise
>> Thank you for your comments. I revised it according to this comment.
- Results section: p value needs should be always reported in the text
>> Thank you for your comments. I revised it according to this comment.
- Lines 170-172: add a comment/discussion to the sentence or delete
>> Thank you for your comments. I deleted this sentence and revised around.
- Lines 173-175: “The study design was retrospective and the groups were not exactly comparable. However, the bias is in favor of the CLH group; therefore, it does not fundamentally affect the results of this study.” Comment further on this sentence
>> Thank you for your comments. I revised it according to this comment.
- The discussion section is too brief and needs to be implemented:
>> Thank you for your comments. I added some comments.
- Comment on the complication rate in this case series vs what is reported in literature
>> Thank you for your comments. The classification of complications is different for each study, so the percentage values that come out as a result are very different. Therefore, it is necessary to check the contents rather than comparing the ratio itself, so a few are added to the discussion.
- Further comment factors for ureteral injury, also depending on surgical route
>> Thank you for your comments. I added some comments.
- Recent references and RCT about this topic must be added:
>> Thank you for your comments. I added RCT below.
o Paraiso, M.F.R.; Ridgeway, B.; Park, A.J.; Park, A.J.; Jelovsek, J.E.; Barber, M.D.; Falcone, T.; Einarsson, J.I. A randomized trial comparing conventional and robotically assisted total laparoscopic hysterectomy. Am. J. Obstet Gynecol. 2013, 208, 368.e1–368.e3687.
o Sarlos, D.; Kots, L.; Stevanovic, N.; von Felten, S.; Schär, G. Robotic compared with conventional laparoscopic hysterectomy: A randomized controlled trial. Obstet. Gynecol. 2012, 120, 604–611. [CrossRef]
o Capozzi VA, Scarpelli E, Armano G, et al. Update of Robotic Surgery in Benign Gynecological Pathology: Systematic Review. Medicina (Kaunas). 2022;58(4):552. Published 2022 Apr 17. doi:10.3390/medicina58040552
o Lönnerfors, C.; Reynisson, P.; Persson, J. A randomized trial comparing vaginal and laparoscopic hysterectomy vs robot-assisted hysterectomy. J. Minim. Invasive Gynecol. 2015, 22, 78–86.
I will be glad to review the manuscript after minor revision and extensive English revision.
Reviewer 3 Report
The study was well-designed, and an equal number of cases were compared with robotic surgery and classical laparoscopic surgery under the same conditions. The groups studied did not differ in average age, uterine weight, or body mass index. The authors objectively presented the limitations and strengths of their study.
However, in the conclusion the authors are somewhat neutral, the reader does not get a clear message of what the actual goal of the study was, so my suggestion is to improve the conclusion somewhat.
Author Response
Thank you for your comments. I revised it according to this comment.